# Sex Differences in Lung Cancer

**DOI:** 10.3390/cancers15123111

**Published:** 2023-06-08

**Authors:** Lauren May, Kathryn Shows, Patrick Nana-Sinkam, Howard Li, Joseph W. Landry

**Affiliations:** 1Department of Human and Molecular Genetics, VCU Institute of Molecular Medicine, Massey Cancer Center, VCU School of Medicine, Richmond, VA 23298, USA; mayl3@vcu.edu; 2Department of Biology, Virginia State University, Petersburg, VA 23806, USA; kshows@vsu.edu; 3Department of Internal Medicine, Division of Pulmonary Disease and Critical Care Medicine, VCU School of Medicine, Richmond, VA 23298, USA; patrick.nana-sinkam@vcuhealth.org (P.N.-S.); howard.li@va.gov (H.L.)

**Keywords:** non-small cell lung cancer, estrogen, androgen, progesterone, smoking, cancer immunology, sex difference

## Abstract

**Simple Summary:**

Lung cancer remains the most commonly diagnosed cancer in the United States, only behind sex-specific cancers such as breast and prostate cancer. While not a sex-specific cancer, lung cancer exhibits sex-specific trends. Males are generally at a higher lifetime risk of developing lung cancer and have a higher mortality than females. There are also differences in therapeutic response between the sexes. As lung cancer is a complex disease, this difference is likely a combination of environmental factors, such as environmental exposures, diet, and smoking status, with inherent biological differences, such as the contribution of sex hormones and differences in immune responses. This narrative review focuses specifically on these biological differences and their contributions to this difference. Gaining a better understanding of the biological reasons behind this sex difference could lead to better treatment and screening decisions in the clinic that take the biological sex of the patient into consideration.

**Abstract:**

Sex disparities in the incidence and mortality of lung cancer have been observed since cancer statistics have been recorded. Social and economic differences contribute to sex disparities in lung cancer incidence and mortality, but evidence suggests that there are also underlying biological differences that contribute to the disparity. This review summarizes biological differences which could contribute to the sex disparity. Sex hormones and other biologically active molecules, tumor cell genetic differences, and differences in the immune system and its response to lung cancer are highlighted. How some of these differences contribute to disparities in the response to therapies, including cytotoxic, targeted, and immuno-therapies, is also discussed. We end the study with a discussion of our perceived future directions to identify the key biological differences which could contribute to sex disparities in lung cancer and how these differences could be therapeutically leveraged to personalize lung cancer treatment to the individual sexes.

## 1. Introduction

Despite significant progress in the diagnosis and therapeutic management of cancers, significant disparities exist, with males suffering from a higher incidence and mortality compared to females worldwide [1,2]. This long-standing disparity in outcomes [3] may be driven by both social determinants and fundamental biological characteristics of the disease. Outside cancers of the reproductive system, there are several cancers including lung, melanoma, urinary tract, and glioblastoma that demonstrate great sex-specific trends in clinical presentation, progression, and mortality. These sex differences, and their implications for cancer treatment strategies, have been a focus of a European Society for Medical Oncology multi-disciplinary workshop highlighting their importance to the cancer community [4]. Lung cancer is the leading cause of cancer death [1]; thus, there is an urgent need to elucidate the underpinnings of disease initiation and progression. As such, a better understanding of the contributing clinical, biological, epigenetic, and genetic drivers of sex differences in lung cancer has the potential to inform risk stratification and screening for early disease as well as treatment strategies that take sex into consideration. In this review, we will describe potential biological bases for the observed sex differences in lung cancer and how such biology could be leveraged to tailor treatment of subjects with lung cancer.

### Sex Differences in Lung Cancer—A Clinical Perspective

In the past, males were disproportionally diagnosed with lung cancer, though this gap has been closing and reversing in certain populations in recent years [5,6,7]. Later diagnoses with more advanced stages of disease are more likely to occur in males [5]. This results in a worse prognosis in males with lung cancer and higher mortality than women despite advances in therapy development, and declining mortality for all cancers (https://www.cdc.gov/cancer/, accessed on 1 May 2023). Sex differences in the survival and mortality of early stage lung cancer are reduced with low-dose computed tomography (LDCT) screening [8,9]; however, current eligibility criteria for screening do not include sex; rather, these are focused on age and smoking history [10]. Lung cancer is a complex disease, and as such, the drivers of the observed sex differences are likely to be multifactorial, including a combination of multiple environmental and biological factors. These factors can include sex-differences in behavior towards the healthcare system and environmental exposures, smoking, diet, sex hormone contributions, and differences in immune responses. Controlling for environmental contributions, age, and smoking history, men remain more susceptible to lung cancer, suggesting that underlying fundamental biological characteristics across sex may be contributing to disease [11]. Despite the confirmation of this sex difference, the biological drivers of this difference remain largely unknown.

## 2. Sex Hormones

Sex hormones and their signaling pathways have long been implicated in the pathogenesis of cancers [12]. Sex hormones with known roles in cancer include prolactin (PRL), luteinizing hormone (LH), follicle-stimulating hormone (FSH), gonadotropin-releasing hormone (GNRH), estrogen (E2), progesterone (P) and testosterone (T) [13]. Many of these hormones have well-documented roles in the cancer biology of reproductive organs including ovaries, endometrium, prostate, and breast cancer [14]. Several sex hormones are produced by lung cancers ectopically [15,16], which contribute to the clinical spectrum of disease (Table 1) [17]. Of these hormones, only exogenous E2, P and T have been documented to regulate lung cancer biology, and their role in the disease remains controversial in the literature.

### 2.1. Estrogen

Estradiol (E2) is the active form of estrogen that has been shown in vitro to induce proliferation of lung cancer cells [48,49]. ERα and ERβ are encoded by the ESR1 and ESR2 genes, respectively, and expressed as full-length proteins with several potential transcript variants [50]. ERβ is normally expressed in lung epithelium and lung fibroblasts, and is present in lung cancer cells. Conversely, ERα is typically not found, or expressed at very low levels, in lung tissues. ERα expression is known to be regulated epigenetically in lung cancer through CpG methylation [51]. One study showed significant methylation in lung cancer tissue in contrast to no methylation in normal tissue [52]. CpG methylation has also been observed in lung cancer cell lines and mouse models of lung cancer [52]. The role of ERα is somewhat controversial; despite overall lower levels of expression in some lung cancers, some studies have shown cytoplasmic ERα in non-small cell lung cancer (NSCLC), potentially due to a transcript variant [53,54].

Signaling through these receptors is initiated through E2 binding to the ER. Studies have shown that, in addition to circulating estrogen and exogenous estrogen from hormone replacement therapy (HRT), E2 can be produced by NSCLC cells via aromatase activity [18]. One study demonstrated that aromatase activity was significantly higher in adenocarcinoma, adenosquamous carcinomas, and bronchoalveolar carcinomas than in normal lung tissues [48]. An additional study demonstrated that lower levels of aromatase expression predicted a better overall survival (OS) in women over 65, as well as in younger women with no smoking history [55]. In addition to their own signaling pathways, ERs can also interact with other cell-surface receptors including epithelial growth factor receptor (EGFR), caveolin, and flotillin [18]. Interaction with these receptors can also promote cell proliferation. ERβ expression has, in general, not been associated with any specific grade of cancer, but has been associated with improved OS. In contrast, ERα expression in the cytoplasm is associated with a higher tumor grade and poorer OS [56,57].

Both ERs and aromatase have been identified as potential therapeutic targets in lung cancer. Fulvestrant, an ER competitive inhibitor, significantly reduced tumor volume of NCI-H23 NSCLC cells in a flank model of ovariectomized nude mice [57]. Erlotinib and gefitinib, both EGFR tyrosine kinase inhibitors, were incapable of controlling tumor growth; however, when combined with fulvestrant, these were superior in controlling tumor growth compared to fulvestrant alone [18]. Aromatase inhibitors such as anastrozole and exemestane alone or in combination with traditional chemotherapies also show significant tumor growth control in lung cancer [48,58].

The role of the estrogen signaling pathway in lung cancer biology is controversial, and as a result, its role in lung cancer remains unclear. Estrogens play important physiological roles in both males and females, though their overall abundance is higher in females [59]. Outside of their role in reproductive signaling, estrogens facilitate a variety of biological processes, from protein and lipid synthesis to modulating immune cell populations [60]. In the context of cancer, the involvement of estrogens is tied to estrogen receptor (ER) status. For example, breast cancers are often ER-positive, and frequently depend on signals from estrogens to grow [61,62]. Likewise, there are suggestions that lung cancers with cytoplasmic, in comparison to nuclear, localization of ER may have a worse prognosis; however, this was not observed universally in all studies [63].

It is unclear if estrogens promote or hinder lung cancer. Several studies have shown a higher risk of lung cancer in women taking HRT [19,20,21], but just as many studies show HRT or oral contraceptives having no effect on overall risk [22,64,65]. Additional studies investigating the interplay between estrogen levels and smoking show a higher risk for developing lung cancer among smokers on HRT [66,67]. Another retrospective study showed that survival was higher in women who were not on HRT at the time of diagnosis, especially in women who had a prior history of smoking [22]. Consistent with estrogens promoting lung cancer, there was a measurable increase in lung cancers in males given estrogens to treat heart disease leading to the termination of the clinical trial [68]. Estrogens can also modulate the effects of other cancer-causing factors, such as genetic mutations due to smoking [69]. The cancer-promoting effects of estrogens are further supported by common ER expression in lung cancers, the ability of estrogen to directly, or indirectly through aromatase expression, stimulate lung cancer growth cell culture studies [49]. These studies strongly suggest that estrogen promotes lung cancer growth in clinical practice and in experimental models.

In contrast, there have been several studies suggesting that there is a protective benefit against lung cancer in women who have not had an oophorectomy. A study of nearly 1000 women demonstrated that women who had a non-natural menopause (i.e., an oophorectomy for medical reasons) were at a much higher risk for developing lung cancer than those who went through a natural menopause [70]. Similarly, the same study also showed that women who entered menopause at a younger age than 45 were at a higher risk of developing lung cancer than their older counterparts [70]. Additionally, a study analyzing patients from the Surveillance, Epidemiology, and End Results (SEER) database showed that women with lung cancer who were premenopausal typically had a more advanced disease at the time of diagnosis and had a higher rate of adenocarcinoma than their postmenopausal counterparts [71]. The same study showed that premenopausal women had similar mortality rates to males in the same age range with lung cancer, but that older postmenopausal women had lower mortality rates than their age-matched males. In addition to estrogens, premenopausal ovaries are known to regulate a variety of bioactive molecules, such as progesterone, testosterone, follicular stimulating hormone (FSH), luteinizing hormone (LH), inhibin b, and anti-mullerian hormone (AMH) [72]. In addition to these, leptin, a dual hormone and cytokine, may be associated with poor outcomes in lung adenocarcinoma and squamous cell carcinoma, though more research needs to be carried out into its potential role [44]. The contrasting effects of ectopic estrogen treatments (i.e., HRT) which generally promote lung cancer, and the effects of sex hormone loss resulting from pre-menopause or ovariectomy, which increase the lung cancer risk of women, could be explained by changes in sex hormones or other bioactive molecules other than estrogen such as progesterone.

### 2.2. Progesterone

Progesterone, while traditionally described as a female sex hormone, is present in both males and females as an important precursor molecule for other endogenous steroids. Progesterone and its role in lung cancer has been less extensively studied compared to estrogen. As with estrogen and ERs, progesterone can be produced by NSCLC and the progesterone receptors (PRs) are expressed in NSCLC [33,73,74,75]. Analysis of PR expression in the patients’ tumor samples did not show any correlation with age, menopausal state, ER positivity, or p53 positivity; however, it was more often found in females with early stage disease and in poorly differentiated NSCLC [33]. Additionally, patients with PR-positive lung cancer had a better OS than those with PR-negative lung cancer [76]. In animal studies, progesterone was able to inhibit the growth of lung cancer cells in a flank model in nude mice, and this inhibition was associated with decreased expression of proliferation marker Ki-67, cell cycle regulators cyclin A and E, as well as an increase in cell cycle [33] (pp. 21, 27). However, studies by other groups show no correlation between PR expression and OS [33,75]. Estrogens and progesterone can also work together, as ER signaling can upregulate PR expression [77]. Despite the disagreement in the literature, studies examining the role of progesterone and PR signaling are ongoing and could provide important information and insights into the role of hormones in lung cancer.

### 2.3. Testosterone

In males, androgen testosterone is the primary sex hormone, and research into its role in cancer has been primarily limited to prostate cancer. Elevated testosterone has been associated with increased lung cancer risk in males in some studies [37,38]. However, other studies did not corroborate a correlation between testosterone and lung cancer risk in men [78]. One study in a mutant *Kras/TP53* mouse showed that males that received E2 required a much lower dose of Adeno-Cre in order to form tumors, suggesting that androgens might potentiate the cancer-promoting effects of estrogen or progesterone [39]. In women, the ovaries and adrenal glands metabolize progesterone into testosterone, leading to the possibility that the high levels of testosterone in men, and in women on HRT, could contribute to the observed sex difference in tumor development and progression [65].

## 3. Genetic Factors

Key genetic differences have been identified in lung cancers from men and women. Women more frequently have lung cancers with a driver mutation such as EGFR, ALK, or KRAS [79,80,81]. Additionally, multiple studies have demonstrated differences in key DNA damage repair pathways in the lung cancers from men compared to women. For example, lung cancers in women often have mutations in p53 and female-associated polymorphisms in the cytochrome P450 gene that affect DNA damage repair efficiency [82]. Female smokers have a significantly higher level of aromatic/hydrophobic DNA adducts in non-tumor lung tissue and have higher expression of *CYP1A1* in lung tissue compared to their male counterparts [83]. Higher levels of *CYP1A1* in females lead to a greater metabolism of polycyclic aromatic hydrocarbons (PAHs) from cigarette smoke into carcinogenic intermediates [83]. Higher levels of polycyclic aromatic hydrocarbon DNA adducts were also found in female smokers compared to male smokers, despite there being lower levels of tobacco carcinogens found in females than males [83]. The increased sensitivity of female smokers to DNA damage could be due, in part, to estrogen signaling and its role in cell proliferation [52]. Additionally, males and females metabolize nicotine and other tobacco carcinogens differently [84]. In general, females have more efficient nicotine clearance due to greater CYP2A6 activity, which can be attributable to higher estrogen levels [85]. Additional studies have focused on the differences between males and females in the expression of the X-linked gastrin-releasing peptide receptor (*GRPR*), which can encourage cell proliferation. One study found that *GRPR* was more highly expressed in female than male non-smokers, and that in smokers, it was found at lower levels of tobacco exposure in women than in men, suggesting that female smokers’ two copies of *GRPR* could contribute to lung cancer susceptibility [86,87].

A detailed meta-analysis of the TCGA datasets concluded that lung adenocarcinoma (LUAD) and lung squamous cell carcinoma (LUSC) have an extensive sex disparity in gene signatures (expression, methylation status and copy number changes) [88]. Sex disparity in mutations in LUAD include a female bias for mutations in MED12, F8, DMD, FAM47A, ABCB5 and male bias for mutations in RBM10, COL21A1, ZNF521, CNTN5, SMG1 and STK11. The significance of these findings is unknown because the products of these genes have a variety of functions, including, but not limited to, transcription regulation, clotting, muscle function, membrane transport. Several significant somatic copy number amplification disparities were observed in LUSC, including 17q11.2, 4q22.1 and a deletion bias at 15q12 to females. Males have several amplifications, including 6q12, 8p11.23, 12p13.33, 17q25.1, 20q11.21, 20q11.21 and Xq28, and a deletion bias at 18q23. Gene set enrichment analysis (GSEA) in LUAD discovers gene sets annotated as androgen response, E2F targets, fatty acid metabolism, G2M checkpoint, glycolysis, Myc targets, and the unfolded protein response as being significantly enriched in females over males. LUSC observes similar enrichments in females for E2F targets, G2M checkpoint and Myc targets. The methylation status of genes in LUSC and LUAD shows a female bias and includes categories annotated as interferon alpha response, TGFβ and TNFα signaling, and apoptosis.

## 4. Environmental Factors and Exposures

### 4.1. Smoking

Cigarette smoking is one of the greatest risk factors for the development of lung cancer. In the past, males were more likely to be smokers than women, though this has decreased in recent years, which is coincident with reductions in smoking rates overall [89,90]. Within the smoking population, males are more likely to develop lung cancer than women, though some studies have suggested that the disease development is different in male smokers compared to female smokers [90,91]. Other studies have shown tobacco exposure as having no effect on risk differences between males and females, possibly due to differences in population sampling and how tobacco exposure is characterized (i.e., inhalation depth, type of cigarette, and tar content) [92]. Concurrent with smoking is secondhand smoke exposure, although there is great difficulty in measuring and characterizing secondhand smoke exposure. One meta-analysis showed that women with smoking spouses had an increased risk of developing lung cancer [93]. However, another study from the UK Million Women Study did not find a link between secondhand smoke exposure and lung cancer development [94].

### 4.2. Additional Environmental Factors

Asbestos, while commonly associated with mesothelioma, also contributes to lung cancer development. Occupational exposures leading to carcinogenesis are more often seen in men, but one study has suggested that non-occupational exposures can put women at a higher risk than men [95]. Many of the studies involving asbestos exposure primarily focus on males. Infections are another source of risk for the development of lung cancer. One study showed that, in a non-smoking population, women were more likely to have HPV-positive lung cancer than men, with follow-up studies showing that women with a history of HPV have increased odds for developing lung cancer as compared to men [96,97].

## 5. Immune Responses

The immune response to lung cancer varies between men and women. A meta-analysis of several publicly available lung cancer gene expression datasets shows that women have increased gene signatures consistent with an acute inflammatory response compared to men [98]. Additional studies of similar datasets showed elevated levels of T-cell dysfunction, inhibitory immune checkpoint molecules, and immune-suppressive cells (MDSCs, Treg) in NSCLC tumors in women compared to men [99]. Single-cell sequencing analysis has shown that tumor-associated macrophages (TAMs) have elevated immunogenicity (elevated IFNγ-producing and antigen-presenting) and are assumed to have greater antitumor activity in cells from tumors of female patients [100]. In male-derived tumor samples, TAMs had gene signatures consistent with being more immunosuppressive.

Sex differences in lung cancer, and many other cancers with a sex bias, could be in part because males and females show marked differences in their immune systems. For example, females have been shown to have immune gene set enrichments as compared to males in the context of lung cancer [101]. The underlying drivers for these differences remain largely unknown. In general, females tend to have an overall more robust innate and adaptive immune response than males [32]. Females also typically harbor higher CD4^+^ T cell counts and produce higher amounts of IFNγ than males in response to infections [102]. While this greater response allows for quicker clearance of infections and development of immunity, it also places females at a greater lifetime risk for the development of autoimmune diseases [103]. In addition to differences in immune responses, there are also differences in circulating immune cell populations between males and females, including greater populations of CD4^+^ T cells and B cells in females, and these differences can shift based on age and race [104,105]. One potential explanation for this difference is the location of multiple genes related to immune function on the X chromosome [106]. Genes that encode for the IL-2 receptor gamma subunit (IL2Rγ), Toll-like receptor (TLR)-7, TLR-8, CD40L, and the forkhead box P3 (FoxP3) are all located on the X chromosome. Because of the higher immune system activity in females, cancer cells in females are often more efficient at evading the immune system and have undergone a stringent immunoediting process to avoid detection by the immune system [107].

Sex hormones are known to have effects on immune responses and cell populations. Estrogen has been shown to have a wide range of effects, from increasing NK cell cytotoxicity in vitro but decreasing the secretion of granzyme B by NK cells in vivo [23,24], to inducing changes in T helper 1/2 (T_H_1/T_H_2) responses [25]. E2 increases the production of human peripheral blood mononuclear cell immunoglobulins via the release of IL-10 [26]. Estrogen has been shown to increase the expression of CD22, SHP-1, and Bcl-2, all of which are important for B cell survival and proliferation [27]. Estrogen levels also have direct effects on the T regulatory population size, which, in turn, can affect T cell activity and proliferation [28]. The CC chemokine receptors CCR5 and CCR1 on CD4^+^ T cells can be stimulated by estrogen [28]. Estrogen can also impact the differentiation and activation of dendritic cells via IFN regulatory factor (IRF)-4 production in myeloid progenitor cells [29].

Estrogen signaling is important in a variety of immune cells, with non-classical ER signaling interacting with the estrogen response elements (EREs) transcription factors NFkB, SP-1, and AP-1 [30,31]. A significant portion of activated genes in female T cells have EREs in their promoters, underlining the important role of estrogen in T cell responses [31]. PD-L1 expression can be modulated by estrogen and by several X-linked microRNAs [32]. Progesterone’s effects on the immune system are less convoluted than estrogen’s; progesterone has an overall anti-inflammatory effect, with decreases in IFNy production in NK cells and inducible nitric oxide synthase (iNOS) in macrophages [34,35,36]. Androgens, such as testosterone, also work to suppress immune cell activity and are anti-inflammatory [108,109]. Testosterone has been shown to reduce immunoglobulin production and reduces the production of IL-6, as well as generally suppressing immune responses [40,41,42]. FSH has been shown to stimulate TNF production in bone marrow granulocytes and macrophages [43]. Leptin, which is generally higher in females, can affect the survival and activation of B and T cells, and it can drive IL-2 and IFNγ production [45]. Prolactin also has effects on the immune system; it has been shown to promote CD4+ and CD8+ T cell differentiation, the release of IFNγ from NK cells, and macrophage activation via prolactin receptor activity [47]. Studies have been performed to characterize the role of the immune system in lung cancer. Lung tumors have been shown to have very low antigen presentation and low co-stimulatory molecule expression, allowing them to escape detection by the immune system [110,111].

Smoking can have significant effects on the immune system. Cigarette smoke can induce MAPK signaling, which can affect NFkB pathway activation [112]. Smoking can reduce the activity of neutrophils and the efficiency of APCs, overall T cell activity, and the amount of circulating immunoglobulins [113,114]. However, studies into sex-specific effects of smoking on the immune system have yet to be performed. These studies could provide key insights into immune cell populations that may contribute to the observed sex differences in lung cancer development and progression.

## 6. Differences in Response to Therapy

### 6.1. Immunotherapies

Immunotherapies, such as immune checkpoint blockade (ICB) therapies, are now the standard of care for a subset of subjects with lung cancer. However, the effects of sex on these therapies in lung cancer patients is an ongoing question in the cancer community. Multiple meta-analyses have been performed on clinical trial data for ICBs, which included lung cancer cohorts; however, many of these analyses fail to take into consideration study heterogeneity and different cancer types. One meta-analysis performed by Botticelli and Coll, which focused on anti-CTLA-4, anti-PD-1, and anti-PD-L1 therapies, showed no significant benefit with immunotherapies regarding OS or progression-free survival (PFS) in males vs. females, including those with lung cancer; however, this study did not account for heterogeneity between trials. This study did not demonstrate any benefits to PDL1 immunotherapy across sex in lung cancer [115]. Wu and Coll investigated the efficacy of CTLA-4 and PD-1 inhibitors vs. other therapies and demonstrated a better PFS and OS in males vs. females; however, these findings were not significant in the NSCLC cohort of the analysis [107]. A third meta-analysis focusing only on NSCLC patients with anti-PD-1 inhibitors (pembrolizumab or nivolumab) vs. chemotherapy showed a clear benefit of anti-PD1 inhibitors (pembrolizumab) in males over females [116]. In the NSCLC cohort of another meta-analysis carried out by Grassadonia et al., PFS was higher in immunotherapy-treated males than in females, and anti-CTLA-4 treatment was associated with longer OS in males [117]. Similar results were also observed Laing et al. in a more recent meta-analysis [118]. In contrast, a meta-analysis carried out by Conforti et al. showed longer OS in women than in men treated with ICBs; however, this study excluded a large number of female patients from their final analysis, making interpretation of the results difficult [119]. Multiple other meta-analyses show no benefit of ICBs as monotherapies or as combinational therapies in males vs. females with lung cancer [120,121]. The lack of agreement on the effects of immunotherapies on males vs. females suggests a need for continued research into this issue and better powered studies to elucidate potential sex differences. Gaining an understanding of which therapies might be more beneficial to a certain sex has implications for clinical practice and choosing adequate therapies for each patient.

### 6.2. Chemotherapies

Some studies have shown that males and females have different responses to chemotherapies, whereas other have not documented any differences [122]. In general, women have more adverse events to therapy than men [123]. In addition, there are well-documented differences in drug kinetics, clearance, and toxicity between men and women [124]. Relevant to lung cancer, platinum-based chemotherapy clinical trials in NSCLC found that women responded better to these chemotherapies than males and had a significantly better OS [125]. However, several studies did not observe these differences with platinum-based compounds [126,127]. In addition, several meta-analyses found that chemotherapy effectiveness in the adjuvant and neoadjuvant setting did not find any sex-difference in the final outcomes [128,129], nor any sex difference in concurrent platinum-based therapy compared to radiotherapy in lung cancer patients [130]. Female patients with lung carcinoma who received paclitaxel combined with carboplatin showed longer PFS than males, which could be potentially explained by a lower activity of DNA damage repair mechanisms in tumors found in females compared to those found in males [131]. Additional studies on combinational therapies with cisplatin have all shown significantly better responses in females over males with NSCLC [132,133]. As such, further investigating sex disparities in lung cancer therapeutic responses could close our knowledge gap as to why they occur and help develop sex-specific treatment strategies to mitigate the detrimental effects of these differences.

### 6.3. Targeted Therapies

Previously, women were shown to have better responses to EGFR inhibitors than males [134]. A recent trial showed a trend of females receiving osimertinib (a third-generation epidermal growth factor receptor tyrosine kinase inhibitor) having improved survival over males; however, this result was not statistically significant [135]. A similar result was observed in a trial studying the use of ALK inhibitors alectinib over crizotinib in patients harboring ALK-mutated NSCLC. Women trended towards improved survival over males, but the finding was not statistically significant [136]. VEGF blockade by bevacizumab showed a higher OS rate in females as compared to males with NSCLC [137]. This result underlines the need for further studies that consider sex as a variable to better understand the different responses to targeted cancer therapies between males and females.

## 7. Models to Study a Sex Difference in Lung Cancer

Model systems will aid in the study of mechanisms driving sex differences in cancer incidence and mortality in the clinic. Invertebrate models including *D. melanogaster* and *C. elegans* may not be useful for this purpose since, to our knowledge, no reports have been published describing a cancer sex difference in these models. In contrast, there are several vertebrate model systems with an observed sex difference in cancer incidence or mortality. In *D. reno*, both Myc and KRasV12 oncogenes preferentially induce liver cancer in males over females, with additional studies showing that androgens promote and estrogens inhibit cancer cell growth [138,139]. Studies in dogs and cats observed a similar sex bias to those observed in humans. Male dogs have an increase in cancers from several origins, including lymphatic, skin, genital, soft tissue, and respiratory system, whereas female dogs have an increase in mammary tumors [140,141]. In general, sprayed and neutered dogs show an increase in some non-reproductive tumors including the heart, prostate, urinary tract, lymphatic and bone cancers [142], suggesting, similarly to humans, that sex hormones influence cancer incidence rates.

A significant body of literature describes sex differences in cancer incidence in mice, including several commonly used inbred strains [143]. The spontaneous cancers in mice are predominantly lymphoma, with a lung cancer incidence at ~5% of mice necropsied at death from old age—[143]. However, in some F1 hybrid strains, it can be as high as 30% with a male bias [144]. Because of this low incidence (~5%), and long latency time (~2 years) a variety of syngeneic, chemical-induced, and genetic models have been developed to study lung cancer. In each of these induced models, a sex difference is observed that mirrors what is observed for humans.

A common syngeneic mouse lung cancer model utilized is the syngeneic Lewis lung carcinoma (LLC) model [145,146]. The use of a subcutaneous model of LLC shows no difference in growth between the sexes in some studies [147,148] or an increased growth in males over females when fed a high fat diet [149]. Ovariectomy causes subcutaneous LLC tumors to grow more quickly in female mice [150,151]. LLC orthotopic tumors grow more quickly in females than in males [152]. The urethane model is a widely used and well-characterized mouse model of carcinogen-induced lung cancer [153]. The urethane model shows a clear difference between male and female mice from several inbred backgrounds, with tumors appearing earlier in male mice than in female mice [154,155]. Ovariectomized, but not castrated, mice develop urethane-induced lung tumors more quickly than their intact counterparts, suggesting that female sex hormones suppress lung cancer growth [156].

Genetically engineered mouse models (GEMMs) have also played a significant role in lung cancer research. A sex difference has been observed in the Kras, p53 (KP model) with some studies showing gender differences in survival [157]. Ovariectomy of KP mice causes tumors to form more quickly and grow more aggressively than their intact counterparts, but administration of exogenous estrogen can also cause greater tumor formation [39]. Administration of estrogen to male KP mice resulted in higher-grade tumor formation than that observed for their vehicle counterparts, indicating an important role for estrogen in this specific model.

## 8. Discussion

While the literature largely agrees that there is an observed sex difference in lung cancer, the underlying mechanisms and drivers have yet to be fully elucidated. The consensus is that men have a higher tumor grade at diagnosis and an increased mortality from lung cancer compared to women. Interestingly, women tend to respond better to cytotoxic and targeted therapies, whereas men have a better response to immunotherapies. It is also clear that there are key differences in the immune systems between men and women, with women having an enhanced innate and adaptive immune response compared to men (Figure 1A). In addition to considerations of sex, racial and ethnic differences must also be taken into account. Data from both the SEER program and from the CDC clearly show higher rates of lung cancer in African Americans than in whites, and those of Hispanic ethnicity had a lower incidence of lung cancer than non-Hispanics [158]. In addition to this, African American patients have lower survival rates than white patients, and African American patients are less likely to receive surgical resection and/or timely treatment compared to white patients [46,159]. Racial and ethnic differences in smoking rates, as well as the impact of other socioeconomic factors, can also have impacts on incidence and survival [160]. In addition to socioeconomic factors, there are also genomic and epigenomic differences between racial and ethnic groups that can influence lung cancer incidence. In one example, a recent analysis of the TCGA datasets identified key gene expression, protein expression, and pathway differences which corelate with differences in LUAD survival between ethnic groups (https://doi.org/10.3390/cancers15102695 (accessed on 1 June 2023)). In other studies, patients of East Asian descent are more likely to have *EGFR* mutations, for example, which could affect treatment and screening decisions for this population in addition to any adjustments for sex [161]. Outside of *EGFR,* there are few studies that consider specific racial and ethnic differences in oncogene mutation frequencies; more work needs to be carried out in this area to identify potential differences that could influence treatment decisions. How these differences could contribute to the clinical differences in lung cancer between men and women is an area of active research.

Key areas need to be the focus of further investigations. Further work needs to be carried out to elucidate the role that estrogen and hormone replacement therapy plays in lung cancer risk in women, as well as to study the roles that progesterone and testosterone may play. The potential effects that hormone replacement therapy may have on the transgender population’s risk for lung cancer must also be considered. Transgender adults in the United States have a higher smoking rate than cisgender adults, putting them at higher risk for smoking-related diseases, and one study showed a high incidence of lung cancer in a population of transgender women after gender affirmation therapy in Amsterdam [162,163]. While research that includes and that is dedicated to the transgender population is still uncommon, more work must be carried out to include this population (Figure 1B). Future studies that consider menopausal status, not just age, and measurements of sex hormone abundance may help clarify the exact functions for menopause/sex hormones in lung cancer. Estrogen’s effects on immune cell function are an area that could be particularly important, as this could not only explain differences in cancer initiation and progression, but also immunotherapy response and potential combinations of immunotherapies with other therapies, potentially aromatase inhibitors that would prevent estrogen production. These categories also need to be considered in therapeutic clinical trials, as estrogen and/or progesterone status may affect therapeutic effectiveness in female patients; for example, sex hormones can affect PD-1 pathway activity, though the effect this may have on immunotherapy response has not been clearly defined (Figure 1B) [164]. In regard to hormone replacement therapy, studies need to be performed on both pre- and post-menopausal individuals that take HRT to assess the potential contribution that it may have to lung cancer development. The potential influences that HRT may have during treatment on both the cancer itself and immune cell behavior need to be studied to potentially develop recommendations for either continuing HRT during treatment or ceasing use. Less frequently discussed hormones such as activins, follicular-stimulating hormone (FSH), and prolactin have all been implicated in other cancer types; however, little work has been carried out as regards lung cancer (Table 1). Activin-A has been shown to be particularly important in melanoma, where it can affect metastatic potential through evasion of the immune system [165], and one study has shown that prolactin may be a possible early predictive factor in lung cancer patients with metastases [166]. It is important to also focus studies on males, not just females. As an example, the male hormone androgen has been recently identified as a key regulator of CD8 T cell exhaustion in models of bladder, melanoma, and colon cancer, but not of lung cancer [167,168]. Ultimately, it will be important to recognize that sex hormones influence a wide variety of processes in both the tumor cells and in the body. Thus, they could be playing both pro-tumor (direct effects of estrogens on lung cancer growth), and antitumor effects through an enhanced antitumor immune response (Figure 1C). Gaining a solid understanding of a mechanistic role they may play in lung cancer development and progression, immune responses to lung cancer, and therapeutic response is necessary for patient wellbeing and the development of new treatments.

While it is clear that the immune system plays an important role in lung cancer development and progression, further studies must be carried out to characterize specific differences in immune responses to lung cancer between males and females. Regarding the immune system, exploring specific pathways that could have different activity between males and females, such as apoptosis-inducing pathways, or checkpoint regulators such as PD-L1, could be a novel avenue to explore for a sex difference. As new immunotherapies continue to be developed and brought into clinical practice, knowing specific pathways that behave differently between males and females can inform not only therapy design but also novel combinations, whether with traditional chemotherapies or newer targeted therapies. Recent work using a TRAIL agonist shows promise in multiple tumor types, though the studies were only performed in female mice, and more studies would need to be performed to study their effectiveness in males [169]. Additionally, androgens have been previously shown to regulate tumor cell checkpoint expression [170,171]. Identifying specific differences in immune cell function and activity between the sexes would open up possibilities for targeted therapies that could take advantage of a pathway with greater activity towards tumor cells from one sex to improve outcomes, or that could try to activate pathways in one sex to improve the antitumor immune response. A better understanding of these differences in the immune system could, in turn, help explain the differences in the response to immunotherapies and traditional chemotherapies.

Studies on the beneficial effects of immunotherapies need to be performed with a potential sex difference in mind; study populations need to intentionally include both sexes and control for menopause in female patents—not just age—in recruitment and analysis. Several analyses have been performed to date with inconclusive and conflicting results, suggesting that not all the variables that affect immunotherapy outcomes have been defined [172]. As new immunotherapies continue to be developed, intentional work needs to be carried out prior to entering the clinic to determine any possible sex differences in response by including both sexes in any animal work and in phase 1 trials. Gaining a better understanding of what therapies are most beneficial for each sex could help guide decision making in clinical practice and ensure each patient receives the best possible treatment, whether that is an immunotherapy, a conventional chemotherapy regimen, a targeted therapy, or a combination. This is especially important as women are often more likely to experience greater toxicity from many traditional cancer treatments [124]. The development and administration of treatments must be equitable between the sexes and must therefore be based on research that considers these factors.

Studies into genetic factors have provided potential targets for therapy; however, more research is needed to continue to study the differences between males and females in DNA damage response pathways and how those can be influenced by smoking. Even though tobacco smoking rates continue to decline worldwide, newer smoking-related activities such as vaping require similar studies into their potential for causing lung cancer and whether there may be a genetic basis for any sex differences. Because vaping is such a new technology, determining its effects will take time. However, there is reason for concern; the analysis of vaping fluids shows high levels of known and probable carcinogens, and the United States may face a significant health burden as the effects of vaping fluid consumption become apparent in the future [173]. In addition to the effects of tobacco consumption and vaping, more studies into other environmental factors and the role they may play in DNA damage and epigenetic changes need to be performed.

Additionally, research needs to be carried out to characterize the contributions that the patient’s microbiome makes to lung cancer development and progression and how this could impact any observed sex differences. The microbiome has well-characterized roles in human cancer [174] and therapy effectiveness, including immunotherapies [175]. The gut microbiome regulates hormone-dependent immune cell activity, suggesting that the microbiome could be a contributing factor for sex differences in the cancer antitumor immune response [176]. In addition to the gut microbiome, lung cancer can be influenced by the lung microbiome [177]. One study has already shown that aberrations in the lung microbiome can increase the risk of malignancy, and another study has shown that treatment with antibiotics during immunotherapy can actually lower progression-free survival—though neither of these studies examined a sex difference [178,179]. The microbiome is rarely, if ever, controlled for in clinical studies and could be a significant variable in outcomes in studies investigating sex disparities. Further research in this area could contribute to a greater understanding of basic sex differences in lung cancer development and progression, as well as differences in immune responses and responses to therapies.

## 9. Conclusions

Lung cancer shows a clear difference in presentation and therapeutic response between the sexes, though specific biological mechanisms that would explain this phenomenon have yet to be described. This difference is likely a combination of environmental factors and innate biological factors working together in the disease process. Further work needs to be carried out to study the role of sex hormones and the role of the immune system in facilitating this difference, as well as to study differences in the therapeutic response and the potential biological reasons behind this difference. Gaining a deeper understanding of the biological reasons behind the sex difference may open the door to new therapeutic combinations or novel therapies based on differing biological mechanisms between the sexes.

## Figures and Tables

**Figure 1 cancers-15-03111-f001:**
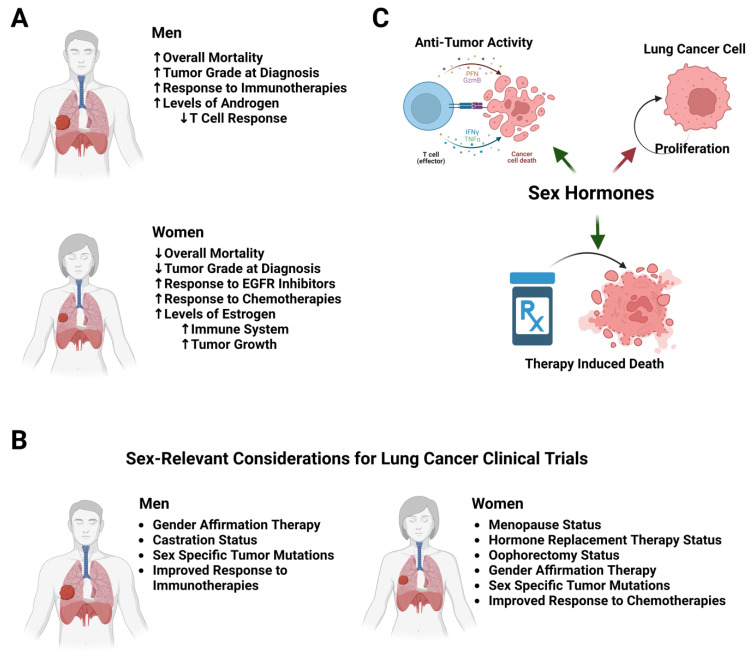
Sex differences in lung cancer growth and treatment. (**A**) There are distinct differences in lung cancer biology between men and women. Men have an overall increase in mortality from lung cancer and an increase in tumor grade at diagnosis compared to women. Due to increases in mutations in the EGFR pathway, women have a better response to EGFR inhibitors than men. Women also have a better response to chemotherapies compared to men, while men have a better response to immunotherapies compared to women. Some of these differences could be due to sex hormones. Estrogens can improve some aspects of the immune response to cancer; however, they can also promote tumor growth. Conversely, androgens suppress the T cell response, possibly contributing to lung cancer in men. (**B**) Several sex-relevant considerations must be accounted for when completing lung cancer clinical trials. These sex-relevant variables can impact treatment outcomes as described in this review. These include a sex-specific mutation status for tumors from men vs. women, the hormone status of both males and females, the inherent differences in therapy outcomes due to sex hormones, and any impacts which could result from gender affirmation therapies. (**C**) Estrogen is the best characterized sex hormone. It has several roles in both promoting and inhibiting lung cancer. Estrogens are a mitogen which promote (red arrow) the cell autonomous proliferation of lung cancer cells, a pro-cancer effect. Estrogens also enhance (green arrow) the cell autonomous response of cancer cells to cytotoxic chemotherapies. Lastly, there is significant literature that shows that the anti-tumor immune response is modulated by estrogen (green arrow), which could explain why women have a lower incidence and mortality for many cancers, not just lung cancer. Figure was created using Biorender.

**Table 1 cancers-15-03111-t001:** Table describing sex hormones and their effects on lung cancer growth. Sex hormones and bioactive molecules have known functions in regulating several aspects of lung cancer biology. These include the cell autonomous effects of lung cancer such as cell proliferation, cell death responses to therapy, and the ability of cancer cells to metastasize. These molecules have significant effects on the immune system, including modulating T cell activity, immunoglobulin production, and innate immune cell responses (cytokine release).

Hormone	Higher in…	Role in Lung Cancer	Effects on Immune System
Estrogen	Females	Can be produced by NSCLC cells [18] Can induce proliferation [18] Through HRT, may increase risk in women [19,20,21] May also exhibit a protective effect depending on age, smoking status [22]	Modulates NK cell cytotoxicity [23,24] Affects T helper responses [25] Increases production of peripheral blood mononuclear cell immunoglobulins via IL-10 [26] Promotes B cell survival and proliferation [27] Affects T regulatory cell populations [28] Can stimulate CD4+ T cell responses [28] Affects dendritic cell differentiation and activation [29] Can interact with estrogen response elements in T cells [30,31] Can modulate PD-L1 expression [32]
Progesterone	Females	Can inhibit growth of lung cancer in vivo [33] PR+ lung cancer may be associated with greater overall survival [33]	Can decrease IFNγ production in NK cells [34,35] Can decrease inducible nitric oxide synthase in macrophages [36] Overall anti-inflammatory [34,35,36]
Testosterone	Males	Elevated levels have been associated with increased risk [37,38] May potentiate cancer promoting effects of estrogen [39]	Can reduce immunoglobulin production [40] Overall suppression of immune responses [41,42]
Follicular Stimulating Hormone	Females	Unknown, though plays important roles in ovarian and breast cancer	Can stimulate TNF production from bone marrow granulocytes and macrophages [43]
Leptin	Females	May be associated with poor prognosis in lung adenocarcinoma and squamous cell carcinoma, though more research is needed [44]	Affects survival and activation of B and T cells [45] Can drive IL-2 and IFNγ production [45]
Prolactin	Females	May be an early predictive factor in patients with metastases [46]	Promotes CD4^+^ and CD8^+^ T cell differentiation [47] Promotes release of IFNγ by NK Cells [47] Can activate macrophages via prolactin receptor activity [47]

## Data Availability

All data used in the paper are publicly available.

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
