# Peer review of "Sex Differences in Lung Cancer"

_cancers, 2023, doi:10.3390/cancers15123111_

Round 1

Reviewer 1 Report

I really enjoyed reading this paper. The authors discuss the evidence for gender differences in lung cancer and then logically and comprehensively discuss biological mechanisms to explain them. These changes are summarised in table 1 where each item is reinforced by at least 1 citation. On completing a discussion of these mechanisms the data is summarised again in Figure 1. This figure is important - it is tweetable and will help disseminate the authors review and highlight the importance of the differences they outline. The figure would be enhanced by coloured arrows in Fig 1 B eg estrogen increases antitumor activity of therapy -green arrow to replace black arrow and reduced cell proliferation red arrow to replace black arrow.

The authors correctly highlight the gender differences found in TCGA - a paper published today Article: TCGA Expression Analyses of 10 Carcinoma Types Reveal Clinically Significant Racial Differences

Brian Lei, Xinyin Jiang and Anjana Saxena Cancers 202315(10), 2695; DOI: 10.3390/cancers15102695

also looks at race - i think one uncharted area is gender-race disparities and their potential importance - the paper above might be considered for cross referencing 

The discussion reviews the implications of their findings eg for clinical trials - i feel another figure (rather than a table)  would help here, again to help make the work more discoverable. Gender and sex are no longer binary - this figure could also highlight our need to be cognisant and accommodating of this in trials and in our studies. At a time when leading candidates for the US Presidency are ghosting this issue it would be important to highlight that we need to be aware of the importance of transgender issues in our research.

references 10, 114 and 122 delete conflict of interest statements associated with the citations

minor edits needed section 2.1 the phrase "are more deadly"- suggest carry a worse prognosis;

p5 of 23 : ovaries can protect against lung cancer - suggest reword this sentence A protective benefit against lung cancer has been reported in women who haven't had an oophorectomy 

p 12 of 23 "in the end " suggest replace with "ultimately"

Author Response

Reviewer #1

Major Concerns

Critique: I really enjoyed reading this paper. The authors discuss the evidence for gender differences in lung cancer and then logically and comprehensively discuss biological mechanisms to explain them. These changes are summarised in table 1 where each item is reinforced by at least 1 citation. On completing a discussion of these mechanisms the data is summarised again in Figure 1. This figure is important - it is tweetable and will help disseminate the authors review and highlight the importance of the differences they outline.

Response: We thank the reviewer for the kind remarks.

Critique: The figure would be enhanced by coloured arrows in Fig 1 B eg estrogen increases antitumor activity of therapy -green arrow to replace black arrow and reduced cell proliferation red arrow to replace black arrow.

Response: We have colored the arrows and have added descriptions of the arrow color in the figure legend.

Critique: The authors correctly highlight the gender differences found in TCGA - a paper published today Article: TCGA Expression Analyses of 10 Carcinoma Types Reveal Clinically Significant Racial Differences Brian Lei, Xinyin Jiang and Anjana Saxena Cancers 2023, 15(10), 2695; DOI: 10.3390/cancers15102695 also looks at race - i think one uncharted area is gender-race disparities and their potential importance - the paper above might be considered for cross referencing

Response: We have added this paper in reference to LUAD into the discussion section of the paper.

Critique: The discussion reviews the implications of their findings eg for clinical trials - i feel another figure (rather than a table)  would help here, again to help make the work more discoverable. Gender and sex are no longer binary - this figure could also highlight our need to be cognisant and accommodating of this in trials and in our studies. At a time when leading candidates for the US Presidency are ghosting this issue it would be important to highlight that we need to be aware of the importance of transgender issues in our research.

Response: We have added Figure 1C to highlight this issue and address these comments.

Critique: references 10, 114 and 122 delete conflict of interest statements associated with the citations

Response: We have deleted the COI statements.

Minor Concerns

Critique: minor edits needed section 2.1 the phrase "are more deadly"- suggest carry a worse prognosis;

Response: We have made the requested changes.

Critique: p5 of 23 : ovaries can protect against lung cancer - suggest reword this sentence A protective benefit against lung cancer has been reported in women who haven't had an oophorectomy

Response: We have made the requested changes.

Critique: p 12 of 23 "in the end " suggest replace with "ultimately"

Response: We have made the requested changes.

Reviewer 2 Report

thank you for asking me to review the manuscript entitled Sex-Differences in Lung Cancer. The authors perform a through review of the topic, analyzing in depth different aspects and summarized the hormonal, biological and immunological differences between sex that could influence tumor growth and response to treatments.

there are 2 points to be addressed:

- is this a review or a narrative review? It seems a narrative one, it should be specified in the title

- It is clear that there are specific differences between male and females in tumor development and response to therapies, but there are also well known differences when considering different etnicyties (eg higher EGFR mutations rate in asian people). Would you consider addressing the combination of this two aspects?

Author Response

Reviewer #2

Major Concerns

thank you for asking me to review the manuscript entitled Sex-Differences in Lung Cancer. The authors perform a through review of the topic, analyzing in depth different aspects and summarized the hormonal, biological and immunological differences between sex that could influence tumor growth and response to treatments.

there are 2 points to be addressed:

Critique: - is this a review or a narrative review? It seems a narrative one, it should be specified in the title

Response: We have made the requested change.

Critique: - It is clear that there are specific differences between male and females in tumor development and response to therapies, but there are also well known differences when considering different etnicyties (eg higher EGFR mutations rate in asian people). Would you consider addressing the combination of this two aspects?

Response: We have added a section to the discussion to address this concern.